# Speech Emotion Recognition Using Attention Model

**DOI:** 10.3390/ijerph20065140

**Published:** 2023-03-14

**Authors:** Jagjeet Singh, Lakshmi Babu Saheer, Oliver Faust

**Affiliations:** School of Computing and Information Science Research, Anglia Ruskin University, Cambridge CB1 1PT, UK

**Keywords:** speech emotion recognition, self-attention models, convolutional neural networks, long short-term memory, RAVDESS, SAVEE, TESS

## Abstract

Speech emotion recognition is an important research topic that can help to maintain and improve public health and contribute towards the ongoing progress of healthcare technology. There have been several advancements in the field of speech emotion recognition systems including the use of deep learning models and new acoustic and temporal features. This paper proposes a self-attention-based deep learning model that was created by combining a two-dimensional Convolutional Neural Network (CNN) and a long short-term memory (LSTM) network. This research builds on the existing literature to identify the best-performing features for this task with extensive experiments on different combinations of spectral and rhythmic information. Mel Frequency Cepstral Coefficients (MFCCs) emerged as the best performing features for this task. The experiments were performed on a customised dataset that was developed as a combination of RAVDESS, SAVEE, and TESS datasets. Eight states of emotions (happy, sad, angry, surprise, disgust, calm, fearful, and neutral) were detected. The proposed attention-based deep learning model achieved an average test accuracy rate of 90%, which is a substantial improvement over established models. Hence, this emotion detection model has the potential to improve automated mental health monitoring.

## 1. Introduction

Research in speech emotion recognition can help in maintaining and improving social relationships and behaviours that are important to survival [1]. Speech is the primary medium of communication for humans and semantic aspects of speech are expressed through the combination of words. In addition, every utterance has a level of emotion attached to it, depending upon the emotional state of the speaker. For example, the way of saying something to inform or declare can be different from asking or persuading that is derived from emotions. Affective speech is characterised by speech signal features, such as bandwidth, pitch, duration, or frequency [2]. Understanding and accurately modelling these variations accounts for most of the research in this field [3,4]. Automated speech emotion recognition can help in many real-time public health applications by recognising or detecting feelings and deducing helpful information regarding the emotional and mental state of patients [5].

The artificial intelligence (AI) community has been actively pursuing research in the field of Speech Emotion recognition. For example, researchers have developed applications for household robots using speech emotion recognition (SER) [6]. The resulting robots could improve their task results by interacting with their owners. There have been several attempts to develop models for SER systems to be used in real-time applications such as: call centres to check customer satisfaction, prioritise songs in song recommendation systems based on the emotional condition of the speaker, and activation of security systems in cars based on the driver emotion level. However, the most important application is related to monitoring the emotional and mental well-being of vulnerable citizens. The mental well-being of citizens has been of utmost importance in the past couple of years due to the mental health problems created by social isolation and remote work culture that was introduced as a result of the pandemic lockdown restrictions [7,8,9]. Currently, physicians place great emphasis on monitoring the emotional sanity of healthy community members, including healthcare professionals [10]. It is tedious and sometimes even hard for humans to decipher and keep track of emotions while interacting with another person. An automated system that monitors the emotional state of interacting individuals can benefit social and healthcare systems by detecting mental health issues. The monitoring can be performed over a large population within realistic time frames that allow for effective remedial actions.

In the last decade, statistical machine learning models, such as Support Vector Machines (SVMs), were commonly used in this research field and they are still a vital part of many research studies [11,12,13]. At the beginning of the 21st century, Hidden Markov Models [14,15] were also used to explore this field of research. However, these methods can only deal with low-dimensional data. Therefore, a dimension reduction step is required that adjusts high-dimensional input signals to low-dimensional data for classification [16]. This dimension reduction can be achieved with a wide range of information extraction methods; a decision on which method to use must be reached during design time. Unfortunately, resolving this choice results in information loss [16]. Information loss is one of the reasons why statistical machine learning algorithms underperform for heterogeneous problems, such as speech emotion recognition. To address this problem, researchers started experimenting with deep learning models, such as Recurrent Neural Networks (RNNs) [17,18,19], which can handle high-dimensional data and therefore do not require feature engineering. Most of these works deal with a limited set of emotions from a single dataset and they are still unable to achieve the desired level of performance on a variety of emotions. The need to improve detection accuracy in this field with a range of emotions has motivated this research to propose novel techniques to be experimented on a combination of datasets to achieve better performance. The hypothesis is that more advanced techniques, such as Convolutional Neural Networks (CNNs) and attention models, can distinguish between subtle emotions. These models should be trained on multiple datasets to improve both performance and robustness. The recent research [20] shows that the use of RNNs, such as LSTM with a directional self-attention layer, could perform well in comparison with established deep learning methods. Attention-based CNN models [21] have shown good performance on certain datasets (accuracy of 77.8% on RAVDESS and 76.18% on IEMOCAP).

In this study, we build on these advanced models by proposing an automated emotion recognition system that tries to decipher a variety of human emotions by analysing the verbal cues from a speaker. Due to the privacy concerns involved in speech recognition, the proposed system does not try to decipher the semantic content of the speech, but only its emotional content based on the way of speaking. This was accomplished with a novel speech emotion recognition system that can be used in healthcare as an automated psychometric analyser based on the recognition of eight different emotions. The system incorporates a combination of CNN and Long Short-term Memory (LSTM) with a self-attention layer. Exhaustive experiments were performed to identify the best combination of features before looking at the best-performing model on the state-of-the-art datasets: Ryerson Audio–Visual Database of Emotional Speech and Song (RAVDESS), Toronto emotional speech set (TESS), and Surrey Audio–Visual Expressed Emotions (SAVEE) independently along with a custom combination of the three. The experiments are also performed using different model combinations such as one-dimensional CNN (CNN 1D), two-dimensional CNN (CNN 2D), and CNN 2D with LSTM to verify the impact of models on task performance along with techniques such as augmentation and normalisation also being applied to this problem to further improve the performance. The proposed speech emotion recognition method contributes to scientific knowledge by:A novel approach that is distinct from methods reported in the standard literature [20,21].Incorporating multiple data sources during the design and evaluation of the speech emotion detection method to increase the portfolio of emotions and variation in data samples.Experiments with different sets of acoustic features for selecting the best combination for modelling.Demonstrating how an increase in data bandwidth could generate better performing models.Delivering a higher accuracy percentage than all the standard models reviewed in this paper.The fact that, to the best of our knowledge, we are the first group who combined LSTM, Attention methods, and CNN-2D to emotion recognition from multiple datasets.

This paper is organised as follows. Section 2 provides the background review of various methodologies and models available in the literature for the speech emotion recognition task. The different datasets, designs, and methodology used for this work are explained in Section 3. Section 4 discusses the experiments and evaluations performed on the model using various datasets along with the experiments performed for feature selection. Section 5 presents the result and comparison of the results with the state of the art along with other observations. Finally, Section 6 concludes the work and scopes the future research in this field.

## 2. Literature Review

Recognising and synthesising emotions in speech have been key research areas over several decades and the associated knowledge has become even more important in recent times due to the popularity of virtual assistants (such as Siri, Alexa, or Google Home) along with its applications in the healthcare sector. Over the last several decades, there has been an emphasis on emotion representation in speech. Montero et al. [22] put forward that synthesised speech cannot be marked as natural sounding in the absence of emotional features. Li and Zhao [23] used acoustic features to identify emotions in speech. They used features extracted from short- and long-term frames of utterances achieving an accuracy of 62% using Gaussian mixture models. In the study from Kandali et al. [24], the researchers used a Hidden Markov Model (HMM) and SVM to classify five different types of emotions. HMMs were used to model the sequential forward selection by identifying the best set of features. The experiments, performed on a Danish emotional speech dataset to establish gender independent predictions, recorded an accuracy rate of 88.9%. In another study by Kandali et al. [25], the researchers recorded emotionally biased speech from 27 different speakers that included some short and long emotional speech from each speaker. It was found that features, such as log-energy, MFCC, and delta-MFCC, performed well in most cases and a Gaussian Mixture Model (GMM) classifier achieved an accuracy of 76.5%. Shen et al. [26] used SVMs on the Berlin dataset with a combination of features, such as energy, pitch, and MFCC to record an accuracy of 82.5%. A group led by Lalitha et al. [27] used SVMs for classification along with a set of multiple spectral features, such as discrete wavelet transform, MFCC, zero crossing rate, and other spectral features. Classical machine learning techniques, such as SVMs, are still popular on this task due to their ease of modelling and evaluation. In a more recent study, Aljuhani et al. [28] used SVMs on an Arabic dataset which recorded an accuracy rate of 77.14% for four emotion states on the dataset created from random YouTube videos following Arabic dialect.

In recent years, Deep Neural Network (DNN) models performed well in many fields, including speech emotion recognition [29,30,31]. CNN 1D was used and experimented on multiple combinations of features for three datasets (Emo-DB, RAVDESS, and IEMOCAP) [32]. An accuracy rate of 85%, 63%, and 76%, respectively, was recorded for the three datasets using LogMel features. Another study on emotion recognition [33] used one-dimensional Dilated CNN (DCNN) with IEMOCAP and EMO-DB datasets and achieved a recognition rate of 73% and 90%, respectively. This work used a simple architecture of residual blocks in combination with a main classification algorithm to find a correlation between emotional cues and sequence of learning. In another study [34], the researchers implemented speech emotion recognition in the Internet of Things using CNN-2D for classification. Data normalisation and augmentation was implemented to accurately classify the data and recorded an accuracy of 95%, which was noted to be the highest as compared to the state of the art.

Since their inception, RNNs have been widely used for speech emotion classification tasks. There are various studies where RNNs have been used in combination with other classification algorithms [35,36]. Pandey et al. [37] analysed multiple combinations of the standard set of speech features such as Mel spectrogram, MFCC, and raw spectrogram magnitudes for EMO-DB and IEMOCAP datasets to identify the combination of features that CNN and LSTM can fit better. It was observed that the MFCC feature provided the best performance and reported high accuracy rates. Parry et al. [38] used cross-corpus training during the model creation. The combination of CNN and LSTM on SAVEE, RAVDESS, and TESS datasets reported an accuracy rate of 72.66%, 53.08%, and 49.48%, respectively, for the three datasets. In their study, Zhao et al. [39] constructed four neural networks to perform a comparative study: two CNNs and two LSTM based models (1D CNN LSTM and 2D CNN LSTM model). The CNN LSTM models were configured to work on the long-term dependencies and, through these experiments, it was recorded that the 2D CNN LSTM network outperformed its counterparts and produced an accuracy rate of 95% on the Berlin dataset for speaker independent data. The other models also performed well with 90%, 89%, and 70%, respectively.

Dolka et al. [40] experimented using a simple artificial neural network model with MFCC as features. The RAVDESS, CREMA, TESS, and SAVEE datasets were used to train and test the models. In a similar study, Asiya and Kiran [41] used a combination of a RAVDESS and TESS dataset for training and testing the model using CNNs and they advocated the use of larger data through data augmentation, because this increases the accuracy. An accuracy rate of 89% was reported using five different spectral features. A language independent speech emotion recognition system was designed using a custom dataset that was created by combining RAVDESS and TESS [42]. For that dataset, an accuracy rate of 78.28% was achieved. It was recorded that the most predictable emotion is anger and most mis-classified emotion was fear using the Mel-spectrogram and MFCC features.

More recently, attention mechanism has gained popularity due to its applications to different learning problems. In a recent study, Xu et al. [43] combined a multi-head-self-attention-based method with a CNN to train models using the IEMOCAP dataset. An accuracy rate of 76.36% was recorded using the spectral features. In another study, Yoon et al. [44] proposed the use of two bi-directional LSTMs to identify the hidden representations of speech signals using a multi-hop attention mechanism that automatically infers the relation between modalities. Peng et al. [45] combined CNN with sliding recurrent networks with attention-based auditory heads to mimic the human auditory system that can easily identify the speaker’s intentions by checking both the intensity and frequency of any utterance. Through various experiments on the IEMOCAP and MSP-IMPROV datasets, the researchers noted that this model can effectively be used to identify the state from temporal modulation cues. They recorded an accuracy rate of 62.6% for IEMOCAP and 55.7% for the MSP-IMPROV datasets. In another study, Chopra et al. [46] performed experiments to check the various combinations of datasets and models and they proposed a meta-learning approach for identifying the emotion.

Many multi-modal speech emotion recognition studies are conducted in the past where researchers also considered other aspects related to emotions such as visual expressions and textual attributes. In general, from an audio clip, the emotional state can be identified based on two factors: the way something is being said that constitutes attributes such as pitch, energy, frequency, and the words used by the speaker. The identification of the words used as clear text helps the model to understand the content and context better. The study in [47] proposed a deep dual recurrent neural network to simultaneously work with text and audio signals using the IEMOCAP dataset and recorded a competitive accuracy rate of 70%. To further explore this concept of multi-modal channels to identify emotion, Khalil et al. [3] studied different models and techniques used for emotional identification where multi-modal channels have been used. In a paper by Parthasarathy et al. [48], the researchers proposed a semi-supervised learning ladder neural network and emphasised the importance of regularizing the model through multi-task learning where auxiliary tasks are learned with the primary task of classifying the emotions. This study recorded a relative gain concordance correlation coefficient between 3.0% and 3.5% for corpus evaluations and between 16.1% and 74.1% for cross-corpus evaluations.

The literature review indicates that scientists have investigated different aspects of speech emotion recognition using a wide range of methods. However, relatively speaking, there are only a few published research studies that used attention mechanisms. In this paper, we document the use of audio signals to explore the feasibility of attention-based neural networks for speech emotion recognition. We compared our results with traditional RNNs and CNNs on customised as well as independent datasets. In addition, feature selection was carried out by experimenting with eight different feature combinations along with using techniques such as augmentation and normalisation. The proposed solutions were entirely based on experimental studies and they can augment future studies in this area.

## 3. Materials and Methods

This project follows a quantitative strategy to deal with the problem. After collecting data from different resources, experimental feature selection was employed using different techniques to come up with the best solution to quantify the accuracy on the datasets as compared to previous works. Figure 1 reflects the overall flow of the proposed model together with the techniques used. First, conducting experiments using selected models on individual datasets, this research then customised and combined the datasets from multiple sources to create an extensive dataset with varied characteristics. Further, the visualisations were generated to understand the characteristics of this combined dataset, followed by data pre-processing to analyse the different properties. Normalisation and augmentation was applied to avoid over-fitting before feature extraction was carried out. The model was constructed with four local feature-learning blocks followed by one long short-term memory block to learn the long-term dependence features followed by an attention layer.

The proposed classification algorithm, which combined CNN, LSTM, and the attention layer, was trained and tested using the aforementioned dataset.

### 3.1. Data Source

There are multiple open-source databases available online for speech emotion recognition problems. As part of this study, we looked at three standard datasets: TESS [49], SAVEE [50], and RAVDESS [51]. These datasets are quite large and they have videos to identify the emotional features from visual expressions. However, only the audio parts were used in this research to create the customised dataset for training and testing the model (after experimenting using individual datasets).

TESS—This dataset contains recorded audio utterances for seven different emotions: angry, disgust, fear, happy, neutral, sad, and pleasant surprise (changed to surprise, to map with other datasets). There are 2800 stimuli in total recorded by two individuals using 200 words in the same carrier phrase.SAVEE—This dataset was also recorded for seven states of emotion recorded by four native English speakers with 120 recordings per speaker. Resulting in a total of around 480 utterances recorded by four male speakers.RAVDESS—This dataset is well-known in the area emotional classification. Therefore, it can be considered as a foundation dataset. It has the same emotional states as the other two datasets except for one extra label for the calm emotional state. It was recorded by 24 actors including males and females. It has additional characteristics, such as modality, vocal channel, and emotional intensity. Only audio files (1920 .wav files) were used for this study ignoring other modalities, such as video.

#### Analysis and Visualisation of Data

From the experiments performed on the individual datasets, it was clear that it is preferable to have more data than any of these individual sets can offer. So, we created a new dataset by combining all three aforementioned datasets. This approach results in 5160 audio files which, together with data augmentation, were used to train and test the model.

Figure 2 demonstrates the general attributes of any emotion. It could be negative or positive, including how being sad and calm can both have low pitches and energy even though one is negative and the other is positive. In the same way, happy is positive and angry is negative. However, both can have higher energy and pitch. Overall, eight emotions (happy, sad, angry, surprise, disgust, calm, fearful, and neutral) were tested in this research. The visualisation of the different types of emotional utterances from all the datasets shows differences in the frequencies between an angry utterance with respect to neutral and calm. In some cases, the spectrogram images have the frequency rate, which is higher for angry and happy emotions in different frames.

Figure 3 shows both waveform and spectrogram for the angry emotion, depicting how emotion changes the frequencies.

### 3.2. Feature Extraction

Selecting the most appropriate features is key to any modelling problem. In speech, there are various types of features, spectral and rhythmic features being the most prominent. Local, global, or both feature types can be extracted and presented to the model. Mel-frequency Cepstral Coefficients (MFCCs) are known to be one of the most efficient features for speech [52]. The experiments performed for feature selection are discussed in the next section.

As shown in Figure 4, MFCCs are the short-term power spectrum of a sound that are developed from adopting a linear cosine transform of a non-linear Mel-frequency scale that was extracted from the sound signals. The signal must be broken into overlapping frames. Once there is a set of continued frames, the fast Fourier transform is applied on it and the Mel-scale is applied. Finally, the cepstral coefficients are generated.
(1)$$MeL\left(f\right)=2595\log\left(1+\frac{f}{700}\right)$$

In [53], an extensive study was performed to compare the neural networks for emotion recognition based on different methodologies, algorithms, and feature-sets. These models used different types of spectral as well as other features. The most successful results were obtained using MFCC alone or in combination with other spectral features. The experimental evidence suggests that spectral features without MFCC or Mel downgrade the overall accuracy. However, combining Mel or MFCC to any spectral feature significantly improves the accuracy. On the other hand, the addition of rhythmic features or using only rhythmic features are not significant in improving the accuracy as shown through experiments and evidence in the next sections.

After extracting the required data features, a normalisation function was applied to normalise the data in terms of the mean and standard deviation. In practice, this operation adopted the form of transforming the data around a unit sphere. Further divide the data into training and testing, keeping 75% for training and 25% for testing. Once the training and test datasets were split, the normalisation function was applied. The label encoders were also used to encode the emotional labels from 0 to 7 that are target values for eight different classes of emotions.

### 3.3. Model Implementation

Starting with a CNN-1D baseline model, multiple experiments were performed and, finally, we came up with a model that incorporated CNN-2D and LSTM with attention based self-learning. For many previous works, the researchers have recorded good accuracy with LSTM and basic CNN models. We used a combined dataset, which was created from various open source resources, and created a distribution with different samples.

As shown in Figure 5, the final model was created from four blocks of convolution layers, which were consist for different layers. The data are being fed to the network from an input layer using a custom function to convert the data into a matrix format, which prepares it for the convolution 2D structure. The result is processed by the convolution block whose output is fed to LSTM layer. Further processing is performed in the attention and, thereafter, in one more LSTM layer. Finally, a dense layer of eight neurons, with SoftMax as activation, is used to predict the probability of all emotions. The emotion with the highest probability is the final predicted emotional state. Equation (Equation 2) defines the convolution block functionality [54].
(2)$$\left(f*h\right)\left(t\right)=\sum_{k=-T}^{T}f\left(t\right)\cdot h\left(t-k\right)$$

Here f(t) represents the Kernel function and working on different speech signals, which were augmented and derived from multiple data sources as h(t−k). To reduce the signal complexity, we used batch normalisation, max-pooling, and dropouts.

Input Layer—As when using convolution 2D network, the input is required as a Matrix format so a custom function is defined to convert the extracted MFCC into the matrix format. In the first custom function for data, we segregated MFCC at a sample rate of 44,100 for the audio duration of 2500 milliseconds.Convolution Block 2D—After multiple experiments, we selected four convolutional blocks for learning the co-relation between local features, with each having five layers in it as below: Convolution 2D Layer—This is the first layer in the proposed model; it uses 64-times impulse features with a kernel size of a 3×3 matrix (from the input layer) to extract important information from the features fed.Batch Normalisation—Batch normalisation is helpful when it comes to reducing the covariate shift that occurs in LSTM models and it helps in improving the overall convergence [55]. This led to an increase in the overall efficiency and causes the LSTM to effectively learn where the covariate shift can abstain to do so.Max Pooling—Max pooling is used to reduce the overall dimensionality of the signal [54].Dropout—Due to the large number of model parameters, the four CONV-2D blocks integrated a dropout layer that randomly sets 20% of the input units to zero [56]. It also randomly skips some neurons during training and sets them out of contribution during the forward pass, whereas no updated weights are applied to these sets of neurons during the backward pass. The convolution block output is reshaped before it is fed to the LSTM layer.LSTM—The long short-term memory layer can learn non-linearity in the data along with long-term dependencies in a sequence. It is a type of recurrent neural network that comprises gates within it [57]. The experiments were performed using CNN and LSTM combinations along with CNN, LSTM, and the attention layer and the later was found to be more performant. In the first LSTM layer of this block return sequence is true, which, in turn, will return the last output sequences. Unlike other recurrent neural networks, an LSTM Cell uses internal gates that allow the model to be trained successfully while back propagating without vanishing the gradient problems. Each LSTM cell outputs one hidden state *h* for each input received and by setting the return sequence attribute to True, which can be accessed through the hidden states generated by the layer. The next layer is the attention layer. There is one more LSTM layer after the attention layer, which serves as the dense layer input. The return sequence attribute for that layer is false as all the states are not required, only the last hidden state serves that purpose.

#### Attention Mechanism

The self-attention mechanism was considered here, because it uses *n* inputs and produces *n* outputs. Internally, the inputs interact with each other, and a comparison is performed to evaluate which input should be provided with more attention. The data with the highest attention score is input to the next layers. If the attention mechanism is not applied, the model can suffer, because of degradation, and lose information during the backpropagation. This problem has been solved, to some extent, by the RNN technique of connecting the bottom layers with the upper layers. However, the attention mechanism can also serve that purpose by not removing the redundant speech signal information and adding weights to the one that is most required. Keras [58] provides a sequential self-attention layer that needs to be installed as a separate module. This layer implements the concept of specifically concentrating on the important features and ignoring the remaining parts. Although LSTM can hold long-time dependencies, it tends to become forgetful in many cases and there is no way to assign more importance to some of the signal features that are more important than many others. Chorowski et al. [59] suggested how the attention mechanism can be used to provide some weight to important features. This framework uses an input (*x*) that is pre-processed by the LSTM layer, one which is an encoder that outputs (*h* = $h_{1}$ to $h_{L}$) in a way that is compatible for the attention mechanism to work with. The attention layer, with a tanh activation function, automatically generates an output *y*. At every *i*th step, this system is generating an output $y_{i}$ by paying attention to the elements provided by the encoder layer.
(3)$$\alpha_{i}=Attend\left(s_{i-1},\alpha_{i-1},\hbar \right)$$
(4)$$g_{i}=\sum_{j=1}^{L}\alpha_{i.j}h_{j}$$
(5)$$\equiv y_{i}=Generate\left(s_{i-1},g_{i}\right)$$

Here, *i* is referring to state of the LSTM cell in the proposed model and the final equation is:(6)$$e_{i,j}=\omega^{T}\tanh\left(W_{s_{i-1}}+vh_{j}+b\right)$$

This attention layer equation specifies how it is keeping the related information from the first LSTM layer and ignoring the remaining layers and then feeding it forward to next LSTM cell with the required information. This structure represents the fact that a single speech signal can have multiple segments and many of these can contain emotional information. Furthermore, these segments can have different emotional saturation in a different period. The weights can be added here to the time dimensions obtained from the LSTM layer, which is based on the attention mechanism. As stated in [60], the attention mechanism calculates the attention weight to be applied on the output from the LSTM cells’ target hidden state. This layer helps in identifying the important information and providing it with a weight on and avoiding the interference from redundant information. The final dense layer uses the input from an LSTM block and the SoftMax activation function converts it into a probability percentage of eight different emotions where the highest percentage is the final output.
(7)$$\sigma {\left(z\right)}_{i}=\frac{e^{\beta z_{i}}}{\sum_{j=1}^{k}e^{\beta z_{i}}}$$

A cross-entropy loss function was used to calculate the difference between two probability distributions and the optimizer used was the Keras–Adam.

## 4. Experiments

The task here was to identify the emotions in speech signals using different features and utilising the best one in the proposed model. In the system design, four models were implemented. The first (baseline) model allowed us to perform experiments on different features. The other three experiments were designed to compare both the accuracy and performance. The first of these experiments was based on simple CNN-2D, the second one incorporated a CNN-2D with an LSTM block, and, in the third model, the attention layer was implemented to specifically look into the important information generated by the LSTM cell and dropout the remaining features. The various experiments performed on different levels are presented below. The standard cross-validation technique is used for evaluating the performance of all the models presented in this paper with a held-out test set on automatically shuffled data. The train–test split was 75–25% and, within the 75% train set, a further 25% of the data were used as the validation set to optimize the model parameters. The final results are reported on the unseen held-out test data.

### 4.1. Feature Selection

Feature extraction is the most important phase of any model development, because feeding incorrect features to the model can downgrade the classification performance. There are multiple features in a speech signal; however, the spectral features are considered as a priority. Therefore, we performed some experiments to evaluate the suitability of different networks for that task. The rhythmic features of speech were also considered and tested thoroughly to check whether they can add value to the overall model accuracy. There are different spectral features available in a speech signal and a Librosa library was used to extract such features from the speech utterances. Basically, the spectral features extract frequencies and power attributes of an utterance and the different spectral features tested are explained below:Zero Crossing Rate (ZCR)—This is the frequency at which the signal crosses zero.Spectral Centroid (Spec-Cent)—This can be defined as the ratio of magnitude of the weighted frequency spectrum to the unweighted one [61]. It generally locates the centre of the spectrum by the impression of the brightness of the sample spectrum of speech utterance.Spectral Bandwidth (Spec-Band)—This is the weighted mean of the frequency of all signals in its spectrum.Spectral Roll-off (Spec-Roll)—This provides the frequency range under which the overall spectral energy belongs to and defines where exactly in a speech utterance the main frequency exists.Root–Mean–Square (RMS)—This is the value mean for each frame of the audio signal from its spectrogram.Chroma-Stft (Ch-Stft)—This computes the chromogram by shifting the time frame across the entire speech signal and expresses how the pitch is been represented with in different time levels.Melspectrogram (Mel)—This computes Mel-scaled spectrograms that render out the frequencies above a certain threshold level.Chroma-cens (Ch-Cens)—This computes the chroma energy normalized that, in term, considers the statistics over a large window of deviations in tempo and articulation and can find out the similar attributes in a signal.Spectral-Contrast (Spec-Cont)—This calculates the difference between the high and low levels of a spectrum.There are some rhythmic features, which are also considered in these experiments but are not proved to be that efficient if used without the spectral features.Tempogram (Temp)—This provides the auto-correlation in a mid-level representation of the tempo information in an audio signal and can specify the tempo variation and local pulse in the utterance [62].Fourier-tempogram (F-temp)—This is the short time Fourier transformation of the onset strength of any speech frame.

Using these features, a baseline CNN-1D model was designed to experiment with the custom dataset. The layer structure of the model incorporates four convolution blocks and one max pooling layer followed by a dropout layer in the third block that is then flattened to provide an input to a 32 neurons dense layer followed by a dropout again and a final dense layer with SoftMax with eight categories according to the number of emotional states.

A recent similar study [63] proposes two sets of features and by comparing these with other combinations proved that a minimalist set of acoustic Low-Level Descriptors (LLDs) when extended with seven more LLDs with cepstral parameters provides decent performance. The authors had 88 parameters as the final set of features. The experiments in this study to select the best features are recorded in Table 1. The main idea is to understand the benefits of using rhythmic and other complex features or feature combinations when compared to simple cepstral coefficients. The feature selection task was performed using the Librosa library where the rhythmic features are compared and it is evidenced that they may not add much value towards the overall accuracy improvement. However, they may only contribute to the model complexity. After experimenting on different sets of features, it was concluded that MFCC as a single cepstral feature can be effective instead of combining multiple features that could increase the model complexity and may lead to increased computational cost to optimize the multiple parameters. Therefore, for further experiments towards our proposed model, MFCC was chosen as the only feature.

### 4.2. Augmentation, Data Quantification, and Gender Impact

The experimental evidence suggests that it is effective to add noise data to the training model to obtain higher accuracy instead of processing the model through only clean data [64]. So, further experiments were performed on different types of two-dimensional models with and without applying augmentation. It was found that augmentation was playing a key role in order to increase the overall model accuracy. In this study, the training data were augmented by adding white noise.

From training the model, with and without added noise, we found that the accuracy significantly increases when the training data has noise. Table 2 indicates that the CNN 2D final test accuracy increased by around five percent when the augmentation was applied. In the same way, LSTM and LSTM with an attention mechanism with CNN 2D were also rising in the overall model performance and significantly raising the accuracy. All these models have the same layer structure except for the further addition of the LSTM cell in LSTM and, then, the integration of the attention mechanism into the proposed attention-based CNN with the LSTM cell.

Combining datasets increases the variation in learning data and, therefore, the performance of the deep learning models [65,66]. Table 3 shows that the proposed model is providing an accuracy of 57.50, 74.44, and 99.81% for datasets SAVEE, RAVDESS, and TESS that have 440, 1920, and 2800 speech files, respectively. This demonstrates that an increase in the data samples increases the model’s performance and robustness. The combined average performance of these three datasets is recorded at an accuracy of 90.19%, which is less compared to the best-performing dataset, TESS, mainly due to an increase in the overall variation of samples. Data quantification is vital to train a model and cause it to learn the patterns in data successfully.

Furthermore, as shown in Table 4, the gender distribution across the datasets are varied. The SAVEE dataset has less samples and only male speakers, whilst the TESS dataset has a larger number of samples and only female speakers. The datasets with a higher number of samples are showing better performance in general. Especially, the female speakers are showing better accuracy compared to the male speakers. This could merely be attributed to the higher number of training samples, with TESS being a larger dataset or, in general, females being able to articulate emotions more distinctly compared to subtle differences in male emotional speech. The acoustic characteristics of female speech are often different from male speech, which could cause it to be easier for the model to distinguish between different female speaker emotions [65]. Female speech tends to have a higher pitch and a shorter vocal tract length than male speech, which could help with extracting more distinct features. Therefore, joining these datasets is not only to augment the data, but to increase the portfolio of emotions along with improving the male/female speaker ratio and gender balance. TESS and SAVEE have seven emotions, namely: angry, disgust, fear, happy, neutral, sad, and surprise, while RAVDESS along with these seven has one more emotional level “calm”. Together, they complement and cover this range of emotions whilst increasing the amount of data and variability in the data.

## 5. Results and Discussion

As noted from the experiment results, the data augmentation adds a significant value to the overall accuracy of the model and helps it to train better. The initial four blocks with the combination of different layers learn the local features where LSTM cells are keeping the long-term dependencies. The attention layer helps to weigh the important features and remove the redundant ones, which is leading to the good overall accuracy. Hence, it was found that the model is competitive and shows an overall accuracy of more than 90% on the test data. The integration of the attention mechanism, therefore, proves to be a good fit to work with speech emotion recognition tasks. The categorical cross-entropy loss for both training and testing data is calculated and it was found that the loss for both datasets decreases with epochs. However, the model training continued to decrease the loss and reached as low as 0.30 with an average loss of 0.33. Similarly, it kept on increasing the accuracy and finally provided an average of 90.19% on the validation data.

Table 5 shows the precision, recall, and F1-score recorded for the proposed model on the custom dataset. Table 5 also shows the received macro average of the F1-score as 90 with the same weighted average.

A confusion matrix evaluates the performance of the classifier using the cross-tabulation of predicted and actual labels. Figure 6 shows the class-wise performance of the proposed model on a customised dataset where disgust is the best performing class with more than 94% accuracy; however, angry has lost its performance as it was wrongly classified mostly into disgust (2.03%) and fear (2.42%).

The emotion upon which the model performed worst was sad with 85.69% and sad was wrongly interpreted as neutral in 5.24% of the cases. Figure 7 shows the spectrogram representation for sad and neutral. It can be observed that they are almost identical. Furthermore, the model confused fear (86.29%) with surprise. One explanation for that failure might come from the fact that the spectrograms for fear and surprise were also almost identical.

Figure 8 represents the spectrogram for the best performing class disgust that can be simply differentiated from the above two and, thus, is the model working correctly.

Figure 9, Figure 10 and Figure 11 show the cross-tabulation of actual and predicted classes for individual datasets. The confusion matrix for TESS, the best-performing dataset, shows that the worst-performing emotion class is “Happy”, which is being confused with “Surprise”. However, it can also be observed for RAVDESS that the “Neutral” class is performing worst and is being confused with “Sad” and “Calm”. The SAVEE dataset samples are not performing well and the worst class here is “Surprise” (5.66%); it is marking the signals “Happy” for around 64%. It can be observed from the confusion matrices that, while female speech (TESS dataset) has limited confusion between emotions other than obvious happy and surprise emotions, male speech (SAVEE dataset) has confusion between all the major emotions including neutral with disgust and sad or disgust with sad and fear or even happy with angry. This further evidences that male emotions are more subtle compared to female emotions in audio.

Table 6 shows a comparison of the proposed model with the state-of-the-art research in the field of speech emotion recognition on the same or similar datasets. This table capture details of different machine learning and deep learning models experimented in the past along with the dataset used and the corresponding performance in terms of accuracy.

## 6. Conclusions and Future Work

In this work, an attention-based two dimensional CNN with LSTM cells is proposed based on a comparison with different model architectures. These models were trained on features that were selected after extensive experimentation as presented in Section 4.1. It is deduced that MFCCs are one of the most efficient spectral features for speech recognition tasks. The Mel-spectrogram can also be considered as a competitive feature; however, only if used in combination with other spectral features. From the experiments and results, it is recorded that the rhythmic features are not able to improve the recognition performance of the emotions in the speech signals. The evaluations were performed using the proposed model on a combination of three contrasting emotional datasets that have shown that data augmentation can be an added value towards improved accuracy; however, the goal was not only to augment the data but also to enhance the range of emotional levels, the variation in data samples, and to improve the ratio of male/female speakers in audio files. Similar results were also demonstrated through experiments on individual datasets. This research also proposes an attention-based neural network to identify and distinguish between subtle variations in the features of the speech signal for close emotions. There is a difference in the performance in terms of the overall accuracy if the results of different datasets are compared. However, it is shown that, for some emotions such as anger and disgust, the model is constantly identifying the classes more correctly throughout the different datasets. The attention mechanism is able to focus on features with specific details that can identify the attributes of the speech signal that play a role in classification and leave out the extra details not helping with the task.

The proposed model was verified on three different datasets and their custom combination. However, there is further scope for testing the proposed model with more datasets that have different characteristics. It can also be observed from the literature that multi-modal data for speech emotion recognition could further improve performance along with prosodic features of speech. In a multi-modal setup, the model may need to process and learn from speech utterances data along with other visual aspects such as facial expressions and body movements. This study can be further enhanced to capture the textual data features to be processed alongside other multi-modal features. Research can also be performed around the application of the proposed model with prosodic features and online learning for integration into a semi-supervised learning environment.

## Figures and Tables

**Figure 1 ijerph-20-05140-f001:**
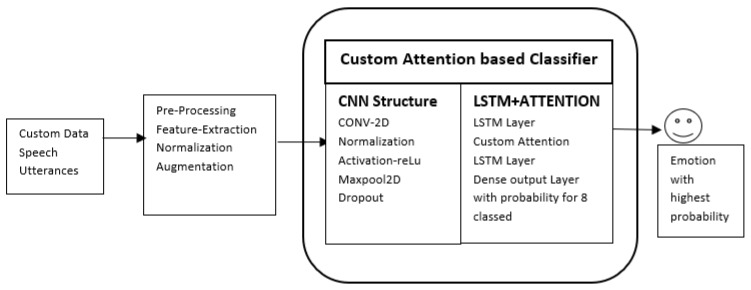
Basic flow diagram of the model and overall process implementation.

**Figure 2 ijerph-20-05140-f002:**
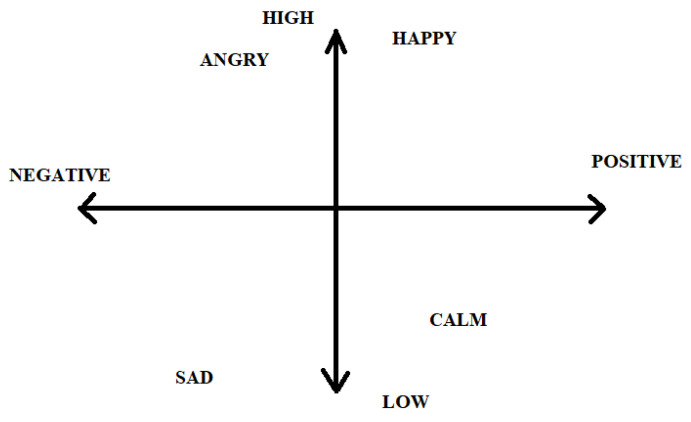
Different emotional levels and variance.

**Figure 3 ijerph-20-05140-f003:**
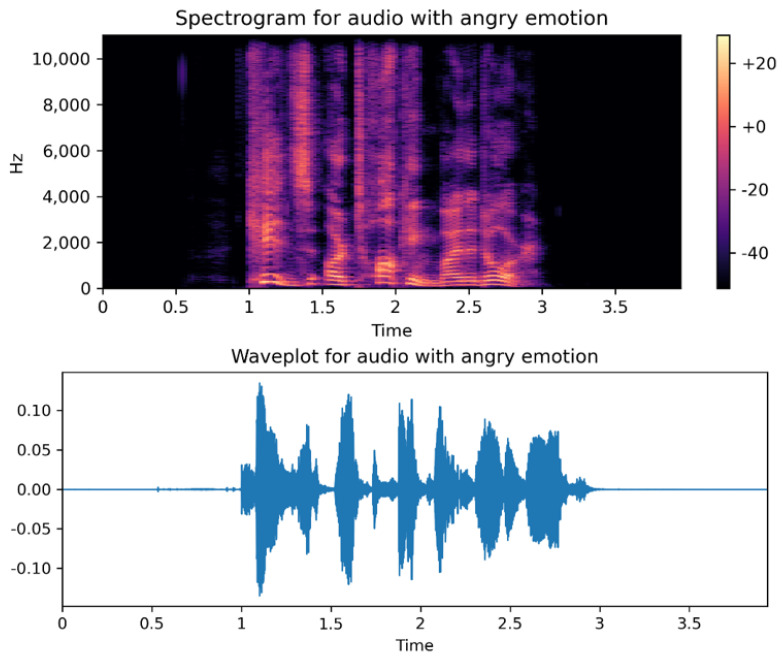
Sample waveplot and spectogram representation for angry emotion.

**Figure 4 ijerph-20-05140-f004:**
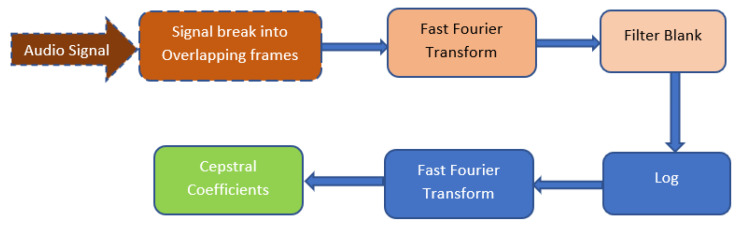
Flow diagram depicting the algorithm chain involved in extracting the cepstral coefficients from audio signals.

**Figure 5 ijerph-20-05140-f005:**
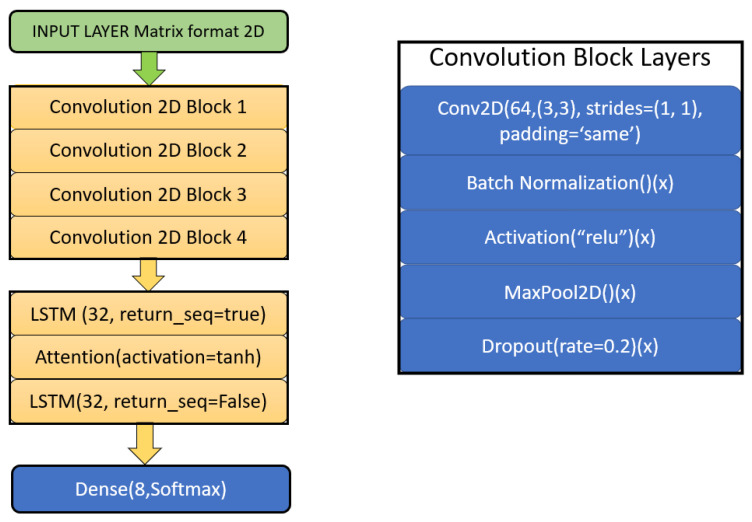
Layer structure of the proposed model.

**Figure 6 ijerph-20-05140-f006:**
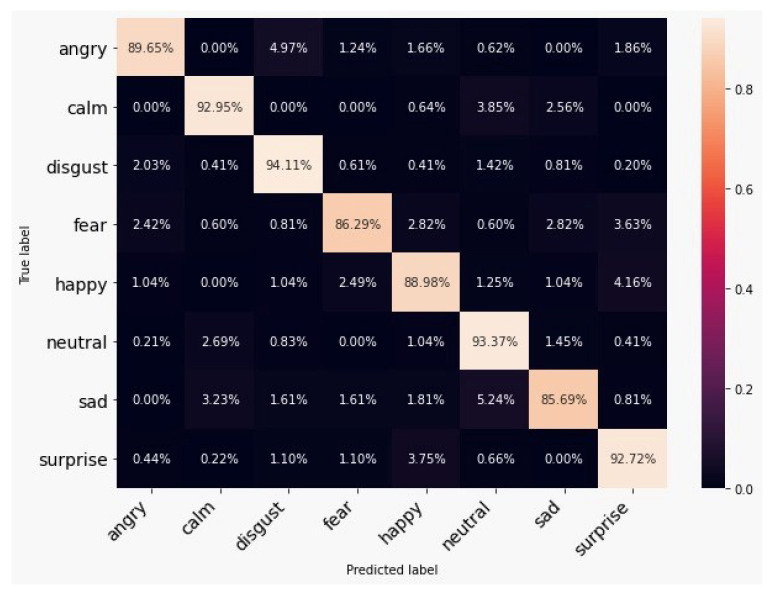
Confusion matrix for the custom dataset.

**Figure 7 ijerph-20-05140-f007:**
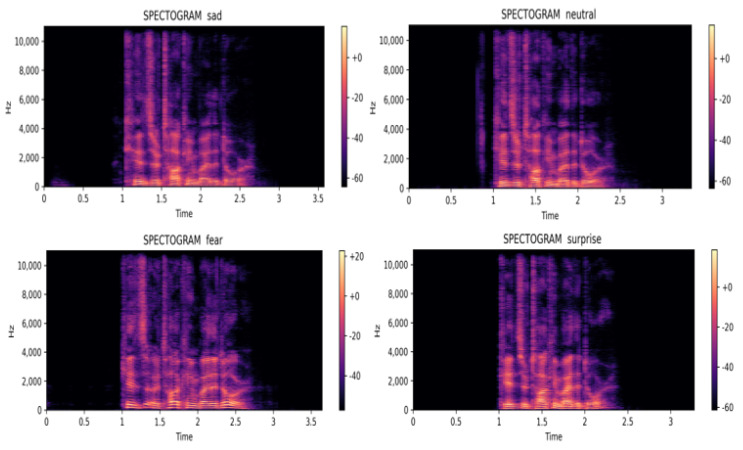
Worst classes and the classes with which worst classes are confused most.

**Figure 8 ijerph-20-05140-f008:**
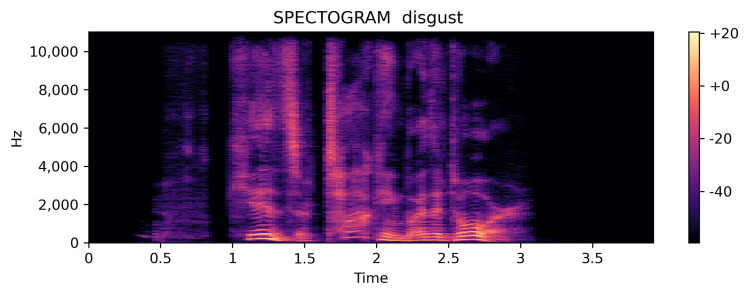
Spectrogram for the best performing class.

**Figure 9 ijerph-20-05140-f009:**
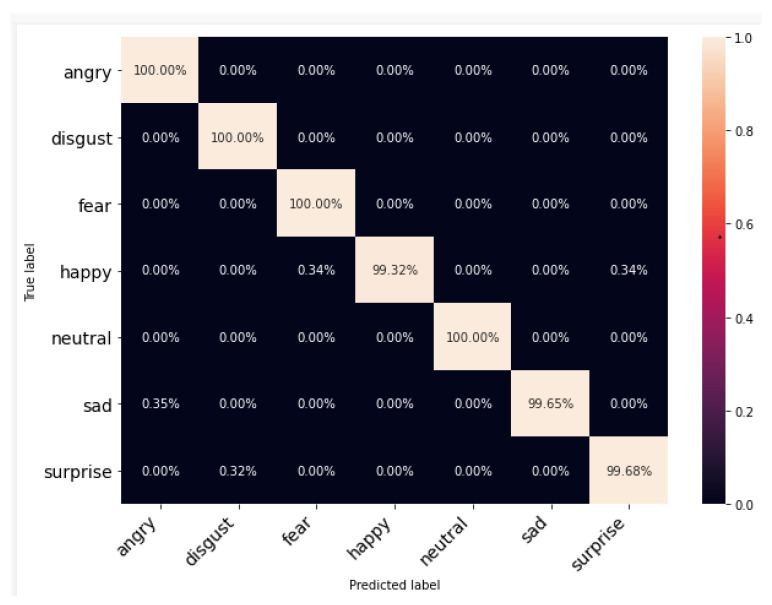
Confusion matrix for each individual dataset using TESS.

**Figure 10 ijerph-20-05140-f010:**
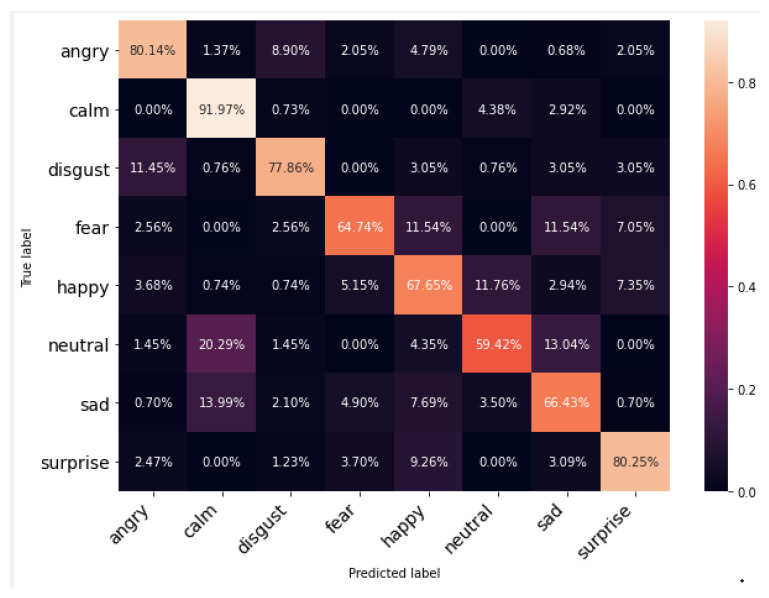
Confusion matrix for each individual dataset using RAVDESS.

**Figure 11 ijerph-20-05140-f011:**
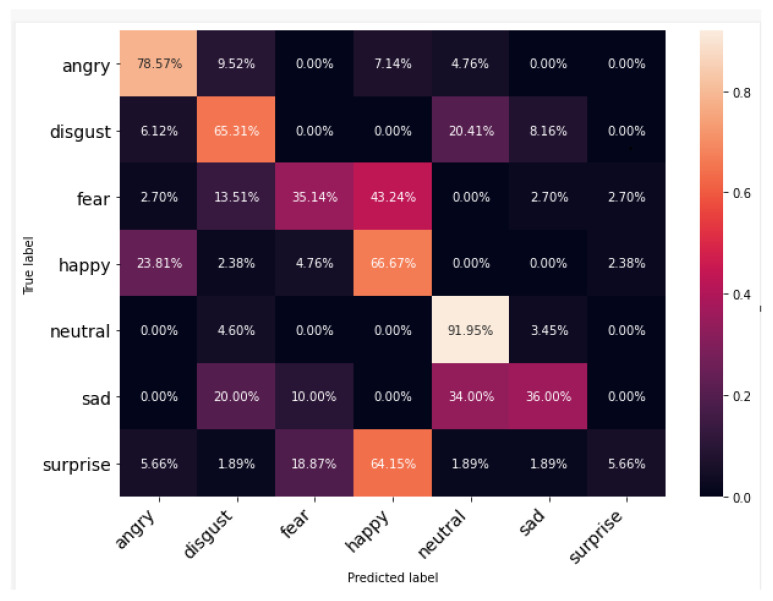
Confusion matrix for each individual dataset using SAVEE.

**Table 1 ijerph-20-05140-t001:** Experiment results on Baseline Model (CNN-1D) for different combinations of features.

SrNo.	Feature Set and Result	Accuracy Percentage
1.1	ZCR, Ch-Stft, MFCC, RMS, Mel	86.00%
1.2	Temp, Ch-cens, Ch-Stft, MFCC, RMS, and Mel	85.70%
1.3	ZCR, Ch-Stft, RMS, and Mel	77.00%
1.4	Tempogram and Fourier-tempogram	13.98%
1.5	MFCC only	85.00%
1.6	Mel only	77.74%
1.7	MFCC and Mel	85.76%
1.8	1.1 + C-Cens, S-Cont, S-Band, and S-Cent	86.00%

**Table 2 ijerph-20-05140-t002:** Experiments on models with and without augmentation using only MFCC.

Model	Augmentation (y/n)	Accuracy Percentage
CNN 2D	No	80.59%
CNN 2D + LSTM	No	81.52%
CNN 2D + LSTM + Attention	No	83.05%
CNN 2D	Yes	85.56%
CNN 2D + LSTM	Yes	86.92%
CNN 2D + LSTM + Attention	Yes	90.19%

**Table 3 ijerph-20-05140-t003:** Experimental results of the three proposed models on individual datasets.

Dataset	CNN-2D	CNN-2D + LSTM	CNN-2D + LSTM + Attention (Proposed)
RAVDESS	73.70%	70.37%	74.44%
SAVEE	60.00%	58.05%	57.50%
TESS	99.71%	99.76%	99.81%
RAV + SAVEE + TESS	85.56%	86.92%	90.19%

**Table 4 ijerph-20-05140-t004:** Distribution of male/female speakers

Dataset	Total Recordings	Male Recordings	Female Recordings
RAVDESS	480	480	0
SAVEE	1920	960	960
TESS	2800	0	2800
RAV + SAVEE + TESS	5200	1440	3760

**Table 5 ijerph-20-05140-t005:** Overall emotional level score metrics.

Emotional Label	Precision (Percentage)	Recall	F1-Score
Angry	0.94	0.90	0.92
Calm	0.81	0.93	0.86
Disgust	0.90	0.94	0.92
Fear	0.93	0.86	0.89
Happy	0.88	0.88	0.89
Neutral	0.89	0.93	0.91
Sad	0.93	0.86	0.89
Surprise	0.89	0.93	0.91
Macro Average	0.89	0.9	0.9
Weighted Average	0.9	0.9	0.9
Accuracy			0.90

**Table 6 ijerph-20-05140-t006:** Accuracy comparison with state-of-art studies.

Method/Model	Reference	Dataset Used	Accuracy Percentage
GMM	Kandali et al. (2008) [25]	Recorded	76.50%
SVM	Shen et al. (2011) [26]	Berlin	82.50%
SVM	Aljuhani et al. (2021) [28]	Custom	77.14%
CNN-1D	Li et al. (2019) [32]	RAVDESS	76%
1D-DCNN	Kwon (2021) [33]	EMO-DB	90%
CNN 1D + LSTM	Basu et al. (2017) [35]	EMO-DB	80%
CNN + Attention	Peng et al. (2020) [45]	IEMO-CAP	76.36%
CNN 3D + Attention	Yoon et al. (2018) [47]	MSP-IMPROV	55.70%
LSTM	Chopra et al. (2021) [46]	TESS, EMODB, SAVEE, RAVDESS	67%
LSTM + Attention + CNN-2D	PROPOSED	RAVDESS	74.44%
LSTM + Attention + CNN-2D	PROPOSED	SAVEE	57.50%
LSTM + Attention + CNN-2D	PROPOSED	TESS	99.81%
LSTM + Attention + CNN-2D	PROPOSED	Customized (RAVDESS + SAVEE + TESS)	90.19%

## Data Availability

https://tspace.library.utoronto.ca/handle/1807/24487 (accessed on 3 March 2022, http://kahlan.eps.surrey.ac.uk/savee/Database.html (accessed on 5 March 2022), https://smartlaboratory.org/ravdess/ (accessed on 5 March 2022).

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
