# Peer review of "Speech Emotion Recognition Using Attention Model"

_ijerph, 2023, doi:10.3390/ijerph20065140_

Round 1
Reviewer 1 Report
1. Searching the combination of the acoustic feature is covered in previous studies, e.g., GeMAPS [a].
[a] The Geneva Minimalistic Acoustic Parameter Set (GeMAPS) for Voice Research and Affective Computing
2. Evaluation settings are not detailed, e.g., cross-validation or dataset split settings.
3. Table 5 is not mentioned in the paper and seems good when it is not included.
4. Using multiple corpora is a simple method and generally improves the accuracies in most deep learning-based tasks and emotion recognition [b, c]. I can't find any novelties in this paper.
[b] Towards speech emotion recognition” in the wild” using aggregated corpora and deep multi-task learning
[c] Multi-Corpus Speech Emotion Recognition for Unseen Corpus Using Corpus-Wise Weights in Classification Loss
5. Using the attention model with CNN and LSTM is general in a range of deep learning-based tasks. This paper does not cover any difference from previous studies.
Reviewer 2 Report
The following comments could improve the quality of paper.
1. On page #2, the authors stated that “the more advanced techniques, like Convolutional Neural Networks (CNNs) and attention models, can distinguish between subtle emotions. These models should be trained on multiple datasets to improve both performance and robustness.” However, the robustness issue is missing in your result discussion.
2. Did you apply K-fold cross validation to all experiments? Please describe in the experimental section.
3. In Table 3, please give more discussion on why the proposed method did not significantly achieve on the individual dataset, especially in the case of SAVEE.
Round 2
Reviewer 1 Report
"automatically shuffled data" will affect the testing performance by the random seed.
Every different random seed yields different splitting, and the trained model. How did you run the testing with multiple trials enough? How did you set the random seed configuration?
Author Response
The authors would like to thank the reviewers for their valuable time taken to review the rebuttal response. To answer the reviewer's query on random seed, all experiments used the same default random seed/split. The data was shuffled and split before experimenting with different models. This ensured that all models used the same data split. There was no effort made to optimise the random seed as this may lead to a biased split. The idea was to use the same data split for all experiments.